# Effective Post-Training Embedding Compression via Temperature Control in Contrastive Training

**Georgiana Dinu, Corey Barrett†,\* Yi Xiang, Miguel Romero Calvo, Anna Currey, Xing Niu**
Amazon, USA
{gddinu,yxxan,miguelrc,ancurrey,xingniu}@amazon.com
†Oracle, USA
corey.barrett@oracle.com

## Abstract

Fixed-size learned representations (dense representations, or embeddings) are widely used in many machine learning applications across language, vision or speech modalities. This paper investigates the role of the temperature parameter in contrastive training for text embeddings. We shed light on the impact this parameter has on the intrinsic dimensionality of the embedding spaces obtained, and show that lower intrinsic dimensionality is further correlated with effective compression of embeddings. We still observe a trade-off between absolute performance and effective compression and we propose temperature aggregation methods which reduce embedding size by an order of magnitude with minimal impact on quality.

## 1 Introduction

Fixed-sized learned representations (a.k.a., dense representations, or embeddings) are the backbone of many machine learning applications across language, vision, audio, and other modalities. Embeddings play an important role in retrieval applications, where given a user input, vector similarity is used to retrieve relevant entries from a previously indexed vector store. This in turn is an essential component of retrieval-augmented generation (RAG; Lewis et al., 2020), which uses retrieval to increase the accuracy and capabilities of generative models. Beyond retrieval, embeddings also have applications in classification tasks, clustering, or parallel corpus mining, to name just a few.

Most embedding models are trained using a form of contrastive loss, either in the entirety of training or after an unsupervised pre-training stage. Contrastive loss functions minimize the distance between an anchor point and other similar data points, while maximizing the distance to a set of negative examples. For training text embedding models, InfoNCE (van den Oord et al., 2018) is the standard contrastive loss used (see equation 1).

The temperature parameter ($\tau$ in equation 1) plays an important role in embedding model training, particularly as it modulates the model's sensitivity to difficult negatives. It has been shown that smaller values of $\tau$ lead to more uniform distribution in the embedding space (i.e., data points are further apart), while larger values lead to lower uniformity but better alignment (i.e., similar data points are closer together) (Wang & Liu, 2020). In parallel, smaller $\tau$ and high uniformity have been found to be correlated with better instance-wise discrimination, whereas lower uniformity and better alignment are correlated with good group-wise discrimination (Kukleva et al., 2023). In this paper, we build on this work, performing an in-depth analysis of the effects of $\tau$ on text embeddings as well as on post-training embedding compression.

**Interaction between temperature and tasks** We start off by investigating the impact of the temperature on different text embedding tasks, where we specifically observe a trade-off between performance on retrieval and on clustering tasks as a function of $\tau$. Diving deeper, we show that

---

\*Work done while at Amazon.

temperature directly impacts the *intrinsic dimensionality* of the embedding space and that lower temperatures lead to greater intrinsic dimensionality.

**Impact of temperature on post-training compression**   These observations further raise the question of whether intrinsic dimension and uniformity properties of embedding spaces impact the ability to efficiently apply post-training compression techniques. We test the hypothesis that smaller intrinsic dimensionality improves quality retention at post-training embedding size reduction (or compression). Embedding compression is essential for reducing costs in practical applications; particularly in retrieval, compression is advantageous because it allows trained models to flexibly adapt to different storage requirements. We show that, as hypothesized, higher temperature does indeed correlate with higher quality retention after compression, and that the correlation holds for *two* different methods of compression: random feature selection or truncation (Kusupati et al., 2024), and quantization via binarization (Bai et al., 2021).

**Techniques for balancing temperature across tasks**   Finally, while these observations shed light on the impact of temperature in contrastive learning, they also confirm the difficulty of training an embedding model that is optimal across multiple tasks and requirements: absolute retrieval results are at odds with quality retention when compressing embeddings. To remedy this, we propose several methods for temperature aggregation and specialization that achieve the best of both worlds; we are able to train models that match the performance of the best single model in retrieval, while showing 99% quality retention when the dimensions are quartered and when the embeddings is binarized.

## 2   PREREQUISITES AND RELATED WORK

**Contrastive losses in representation learning**   Most embedding training algorithms employ a form of contrastive learning, where an anchor point (a word token, sentence, image, etc.) is brought closer to positive examples and sent further away from negative examples. Examples of positive data points are co-occurring words for word embeddings (Mikolov et al., 2013), similar sentences or neighboring sentences for sentence representations (Kiros et al., 2015; Devlin et al., 2019), or noisy versions of the anchor image in vision.[1]

Several variations on contrastive losses have been proposed, most of them inspired by noise contrastive estimation introduced by Mnih & Kavukcuoglu (2013) and Gutmann & Hyvärinen (2012). Here, we use InfoNCE (van den Oord et al., 2018), the contrastive loss most commonly used in embeddings for text and image retrieval. InfoNCE is defined as:

$$L_{\text{InfoNCE}}(f, x, \tau) = \mathbb{E}_{(x,x^+) \sim p_{pos}} \left[ -log \frac{e^{s(f(x), f(x^+))/\tau}}{e^{s(f(x), f(x^+))/\tau} + \sum_{x^-} e^{s(f(x), f(x^-))/\tau}} \right] \qquad (1)$$

where $f$ is the embedding function to be learned, and $x$ is an anchor point compared with a positive $x^+$ and a set of negatives $x^-$ via a similarity function $s$. We choose $s$ to be cosine similarity, which has been repeatedly shown to outperform Euclidean or plain (un-normalized) scalar product for both language and vision applications. This is further scaled by the temperature parameter $1/\tau$.

Many studies have found that the contrastive power of InfoNCE and similar losses increases with the number of negatives and the difficulty of negatives (Lazaridou et al., 2015; Tian et al., 2020; Khosla et al., 2020). In addition to this, it has been shown that InfoNCE loss itself, when used with normalized representations (i.e., using cosine similarity), performs hard negative mining *implicitly* (Khosla et al., 2020). More precisely, it upweights the negative samples that are further away from the anchor in the embedding space. However, whether these are indeed difficult samples or false negatives depends on the supervision signal.

**Impact of temperature**   The temperature parameter is commonly tuned in InfoNCE and related losses in order to obtain better representations. Several studies have shown that temperature further modulates the impact of difficult samples.

---

[1]For a full review of this work, please see surveys such as Kashyap et al. (2024).

Khosla et al. (2020) and Wang & Liu (2020) showed that the difficult negatives have large gradient contributions in InfoNCE. They further investigated the relationship between InfoNCE and other losses when varying $\tau$. When $\tau \to 0^+$, it becomes a triple loss with a margin of 0 using only one negative, the nearest negative:

$$L_{\text{triple}}(f, x) = max[s(f(x), f(x^{*-}) - s(f(x), f(x^+), 0] \tag{2}$$

where $x^{*-}$ is the negative that is maximally similar to the anchor point $x$. When $\tau \to \infty$, InfoNCE converges to the simple loss, which uses all negatives but does not apply a softmax transformation:

$$L_{\text{simple}}(f, x) = -s(f(x), f(x^+)) + \lambda \Sigma_{x^-} s(f(x), f(x^-)) \tag{3}$$

In a similar vein, Robinson et al. (2021a) show that one can upweight the difficult negatives by introducing a new *concentration* parameter $\beta$ in the normalizer of the InfoNCE loss. Empirically, this acts the same as fixing the temperature parameter used for positive samples and varying the temperature used for negatives, where $\beta = 1/\tau$. They show that large $\beta$ leads to upweighting the contribution of the difficult samples, which corroborates the previous results.

One way to visualize this effect is to measure the impact that each negative sample has on the loss normalization factor. The plots in Figure 2 show this impact in two settings: one with random vectors, which is similar to the beginning of training, and a mid-training setting, in which the vectors are drawn from real batches when models are trained with different $\tau$ values (see Section 3 for training details). We use 1024-dimensional embeddings and a batch size of 100 negative samples.

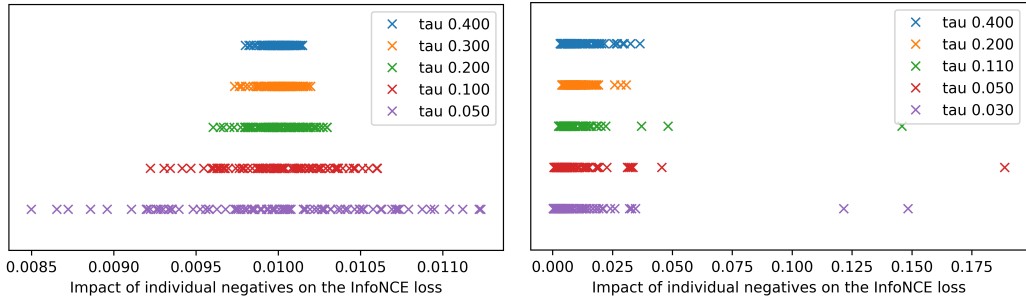

Figure 1: The impact of individual negative samples, computed relative to all negatives in the batch, under different temperature settings. Left: using random vectors; Right: using vectors mid-training.

When the vectors are random the effect is very clear: when $\tau = 0.4$, most data points contribute similarly to the loss, and as $\tau$ decreases, some negatives become outliers, with their contribution being either enhanced or diminished. When the vectors are drawn from real training batches mid-training, the distributions look similar for all $\tau$, with the exception of outliers: with smaller $\tau$ we see that the *difficult* negatives have a larger impact (note that the batch is identical for all models).

**Choosing the optimal temperature parameter**   In computer vision, several works point to performance trade-offs when choosing small vs. large temperature values; despite difficult negatives being preferred in general, a small temperature is not universally superior. Wang & Liu (2020) highlight that while temperature modulates the impact of difficult negatives, there are other ways to do so, such as by explicitly selecting difficult negatives. As such, a large $\tau$ value, which weights all negatives the same way, can be coupled with difficult negatives, leading to better performance overall. Also in vision, Kukleva et al. (2023) show that smaller temperatures are better at modeling tail classes while larger temperatures work best for the head of the distribution. They summarize this as a trade-off between *instance-* and *group*-wise discrimination, where smaller temperatures favor finer-grained instance-level discrimination. As a solution, they propose varying the temperature during training and show optimal results when a large range is used, with $\tau$ from 0.07 to 1.0. Similarly, Robinson et al. (2021b) empirically show that larger temperatures promote the learning of *easy* features, while other features that could provide finer-grained distinctions are ignored.

As a way of explaining these results, several studies (Wang & Liu, 2020; Wang & Isola, 2020) relate their finding to the observation that contrastive losses navigate a trade-off between uniformity and

alignment as defined in Wang & Isola (2020). They show that when the number of negatives $M$ converges to infinity, the normalized loss $(-log\ M)$ converges to:

$$\lim_{M \to \inf} L(f, \tau) - log\ M = -\frac{1}{\tau} \mathbb{E}_{(x,x^+) \sim p_{pos}}[f(x)^T f(x^+)] \qquad \text{Alignment}$$
$$+ \mathbb{E}_{x \sim p_{data}}[log \mathbb{E}_{x^- \sim p_{data}^-}[e^{f(x)^T f(x^-)/\tau}]] \qquad \text{Uniformity}$$

(4)

Alignment measures the similarity of the anchor to the *positive* points, while uniformity measures the similarity to the *negatives*. The goal is to maximize distances between arbitrary points (have a uniform space) while improving alignment.

## 3 TEMPERATURE IN CONTRASTIVE TRAINING OF TEXT EMBEDDINGS

Inspired by the performance trade-offs previously observed in computer vision tasks, we first investigate the impact of temperature in training *text* embeddings.

### 3.1 EXPERIMENTAL SETUP

Text embeddings are typically trained to be able to solve a range of applications, from retrieval to classification and clustering. This is achieved by employing two or more training stages: in a first stage, training is performed on large amounts of raw text using a reconstruction objective such as masked language modeling (MLM; Devlin et al., 2019). This is followed by one or more other stages where supervision consists of a mix of explicit contrastive data, examples including question answering data, semantically similar sentences or documents, documents and their summaries, and so on. We follow the same procedure here, employing two-stage training. We start from a fixed model pre-trained using the MLM objective. We then follow this with InfoNCE contrastive learning; this second stage is the only training stage that we vary in the following experiments.

**Model architecture** We use the CodeSage architecture introduced in Zhang et al. (2024).[2] Our models are trained with 356M parameters and 1024 embedding dimensions. Additional training parameters are given in Appendix A.

**Data** For the MLM training stage, we use 2T tokens of data consisting of 80% English data, with the remaining 20% spanning over 100 languages. We tokenize the text with tiktoken to build a 100k vocabulary[3]. For the contrastive stage, we train on MS Marco (Bajaj et al., 2018; Wang et al., 2023), NQ (Karpukhin et al., 2020; Gao & Callan, 2021), NLI (Gao et al., 2022), HotpotQA (Yang et al., 2018), FEVER (Thorne et al., 2018), MIRACL (Zhang et al., 2023), and Mr. TyDi (Zhang et al., 2021), totaling approximately 2 million data points (see details in Appendix A). We use the training splits of these datasets released by Thakur et al. (2021). We use in-batch negatives and a batch size of 256 data points. Negatives are obtained by randomly sampling other data points from the same training set (homogeneous sampling). Sampling from the same dataset is a common technique used to ensure that the in-batch negatives are not too easy. If a contrastive negative example is explicitly provided with the training data set, that data point it is added to the batch of negatives. Additionally, we assign weights to the negative samples based on similarity to the anchor, following Dong et al. (2024).

**Evaluation** We evaluate using the standard English MTEB benchmark (Muennighoff et al., 2023), which contains a total 56 datasets categorized into eight tasks:[4] classification (12 datasets), clustering (11), pair classification (3), re-ranking (4), retrieval (15), semantic text similarity (STS; 10) and summarization (1). We evaluate on all these tasks but focus on retrieval and clustering, which are evaluated using nDCG@10 and v-measure, respectively.

**Note on MTEB instruction models** It is common in the literature to test embeddings using additional information regarding the downstream task or the text to be embedded, in order to reduce the inherent ambiguity of modeling different tasks (Wang et al., 2024a). Common procedures include using different prefixes for different tasks, or using free-form text (*instructions*; Su et al., 2023;

---

[2]Available at `https://huggingface.co/codesage/codesage-base`.

[3]cl100k_base, available at `https://github.com/openai/tiktoken`.

[4]A leaderboard in maintained at `https://huggingface.co/spaces/mteb/leaderboard`.

Wang et al., 2024b; Muennighoff et al., 2024). However, when used in MTEB evaluations, it is often the case that: 1) instructions are individually tailored for each of the 56 MTEB test sets and 2) test set instructions are paired with the *train* set ones, thus mapping train to test sets. We believe this runs the risk of MTEB over-fitting and thus goes against the original goal of the MTEB benchmark, which is to test robustness across different tasks. For the purposes of this study, we do not use instructions at train or inference time, similarly to models such as Neelakantan et al. (2022).

### 3.2 PERFORMANCE WHEN VARYING TEMPERATURE

We start by measuring the impact of the $\tau$ (temperature) parameter by varying $\tau$ from 0.04 to 0.4 and measuring performance on MTEB.[5]

Results for retrieval and clustering are shown in Figure 2, while results for the other MTEB tasks are provided in Appendix B. We separate retrieval and clustering from the other tasks due to the interesting trend we observe: retrieval performance monotonically decreases with $\tau$ while clustering shows the opposite trend. This result seems to confirm the hypothesis that temperature affects group vs. instance-wise discrimination and that there is a traded-off between the two.

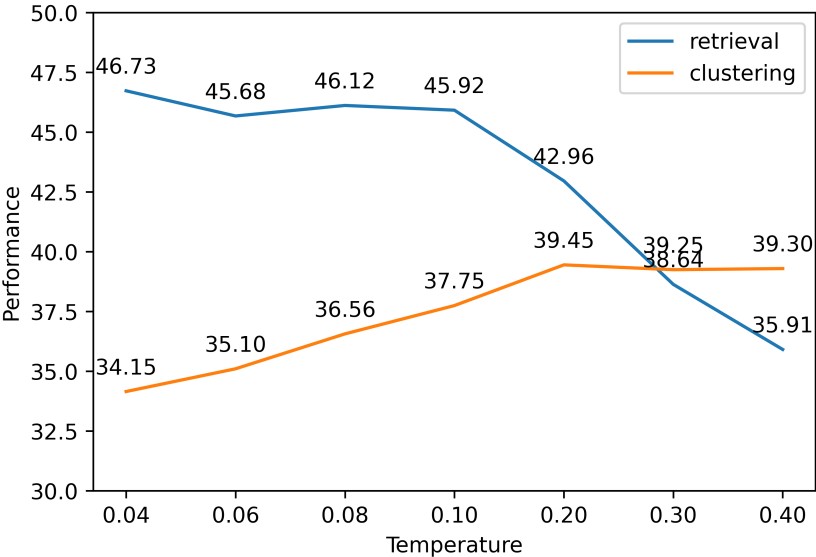

Figure 2: MTEB performance on Retrieval and Clustering tasks, as a function of the temperature value used in contrastive learning.

Building on previous work, we attempt to characterize the distinction between embedding spaces obtained when using different temperature values. For this purpose we use two techniques. We perform t-SNE projection of the embedding spaces by randomly sampling 5000 datapoints and plotting their 2-D representation (van der Maaten & Hinton, 2008). We further implement a uniformity metric similar to the definition in Equation 4: we modify it to use the *normalized* scalar product instead, in order to match our InfoNCE objective which uses cosine. In both cases we sample datapoints from the RedditClustering test set in MTEB; however, results are consistent across other data sets.

t-SNE projections and uniformity values for different temperatures are shown in Figure 3. t-SNE projections indicate the appearance of better defined clusters as $\tau$ increases, which is similar to observations made in previous work for image embeddings. However, the uniformity metric increases when $\tau$ goes from 0.04 to 0.06, but shows a slow downward trend after that. Wang & Liu (2020) also measure uniformity and note that uniformity decreases with large $\tau$; however, the trend does not hold when negatives are selected to be more difficult. Our results also indicate that there is no monotonic relationship between temperature values and uniformity as defined this way. Indeed,

---

[5]We were unable to train strong models with lower temperature values likely due to numerical instability, a phenomenon also reported by Khosla et al. (2020).

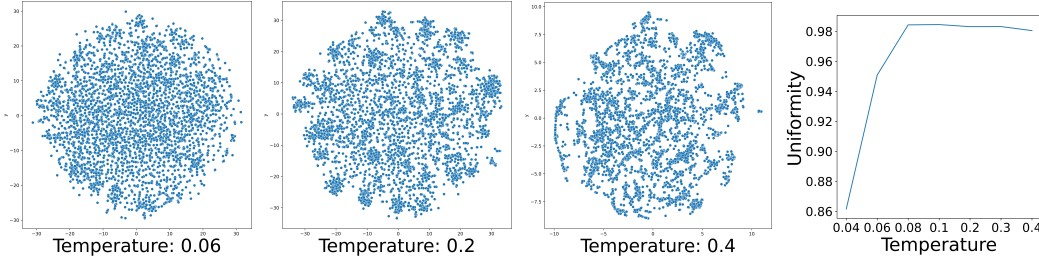

Figure 3: Embeddings projected into 2 dimensions using t-SNE and uniformity values across different temperatures.

uniformity measures mean cosine similarities: if the space becomes more clustered as t-SNE visualizations indicate, this metric can increase or can decrease.

Based on the t-SNE visualization and clustering performance, we hypothesize that the temperature likely modulates the *intrinsic dimensionality* of the resulting embedding space, and a vector space that is more "clusterable" has a lower intrinsic dimensionality. We investigate this by adopting a measure of intrinsic dimensionality based on PCA, which identifies principal components that explain the variance in the data. We perform PCA on the embedding spaces and plot the variance explained by each principal component. Furthermore, we can compute the number of principal components required to explain a certain amount of variance in the data. We use this number as a metric of intrinsic dimensionality and set the variance threshold to 95%.

Figure 4 shows the explained variance of the first 300 principal components under different temperature values and plots the intrinsic dimensionality as defined above. The results indeed show the intrinsic dimension decreasing when larger $\tau$ is employed.

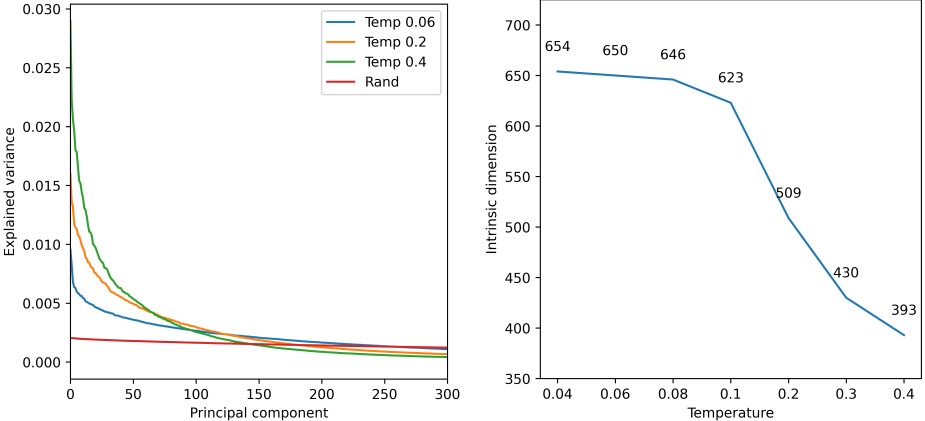

Figure 4: Impact of temperature on the intrinsic dimensionality of the embedding spaces as computed by applying PCA to 5000 datapoints. Left: Explained variance for each principal component. Right: Intrinsic dimensionality calculated as the number of principal components needed to explain 95% of the variance in the data; this number is 896 for random vectors.

## 4 POST-TRAINING EMBEDDING COMPRESSION

The previous section showed that the temperature parameter impacts the intrinsic dimensionality of the resulting space, thus explaining why clustering results improve with large $\tau$. This result brings up a related question: can embeddings with lower intrinsic dimensionality be compressed more efficiently?

**Compressing embeddings** Compression is essential to reduce computational costs associated with storing embeddings for retrieval applications. Furthermore, *flexible* compression, meaning embeddings that perform optimally at full size as well as when compressed, offers even more advantages. When training representations, the computational constraints for each downstream task are often unknown, and in such cases fixed-capacity representations may be too complex or too simple for the task at hand. This section focuses evaluation on retrieval datasets due to the significant costs associated with storing embeddings (Shakir et al., 2024); other tasks are discussed in Appendix B.

For *flexible* compression, simple, computationally efficient operations are preferable: dimensionality reduction via vector truncation (Kusupati et al., 2024) and quantization (Bai et al., 2021; Zafrir et al., 2019) are two such methods to compress embeddings. We investigate random feature selection and binarization (quantization to 1 bit) using the sign function.

**Random feature selection** We expect embedding spaces with low intrinsic dimensionality to retain their quality when dimensionality reduction methods are applied; we hypothesize that this also holds for simple methods such as random feature selection due to the fact that low intrinsic dimensionality implies more redundancy in the feature space. In these experiments, we truncate vectors $\mathbf{x}$ to $\mathbf{x}_{[1:k]}$ with $k \in \{256, 512\}$. This reduces the size of the vectors by a factor of $d/k$ where $d$ is the original size of the embedding space (1024 in our experiments).[6]

**Binarization** We perform post-training binary quantization using the commonly used sign function. This leads to representations that require only $d$ bits for storage. This is a more aggressive compression, reducing storage by 32x assuming 32-bit original precision. For binary embeddings, we replace cosine similarity with Hamming similarity (the number of positions where the two vectors are identical), as recommended by previous work.

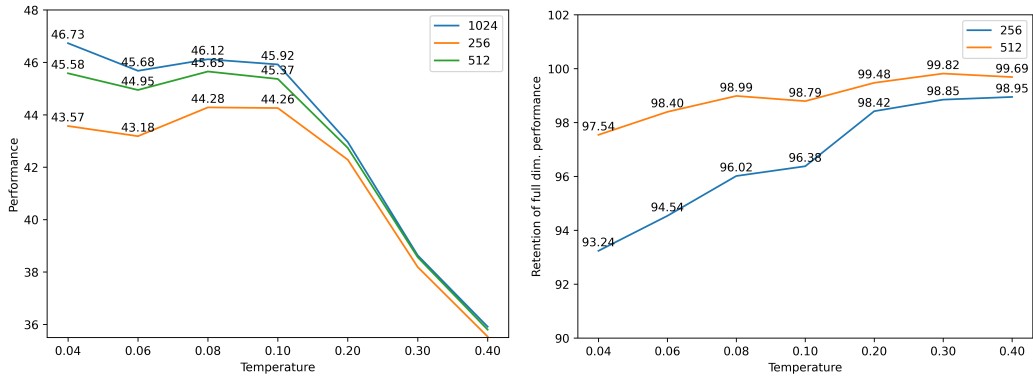

Figure 5: MTEB Retrieval performance when embeddings are *truncated*: Absolute performance (left) and quality retention computed relative to full-size performance (right). In terms of absolute performance, at full dimensionality (1024), retrieval scores decrease as $\tau$ increases. However, after truncation, optimal performance is obtained with larger $\tau$ values of 0.08 and 0.1, due to the increase in *retention* as shown in the right figure.

Results for random feature selection are shown in Figure 5. We show both absolute performance as well as quality retention measured with respect to full size embeddings (1024 dimensions). Larger temperatures lead to increased quality retention when reducing the embedding size. The effect is more pronounced when truncating to 256 dimensions, where the quality retention goes from 93% with $\tau = 0.04$ to 99% for $\tau = 0.4$.

Figure 6 shows a similar trend in the case of binarization, with larger $\tau$ promoting better retention. This is surprising given that lower temperatures lead to embeddings that are sprea more widely. For example, Kukleva et al. (2023) discretize an embedding space into 500 bins and look at the distribution of embeddings across these bins. Lower temperatures show more uniform distribution across these bins, which would have suggested better quantization properties. However, empirically we observe the opposite effect.

---

[6]For the rest of the paper we will continue to the terms truncation and random feature selection interchangeably as they are identical in this work.

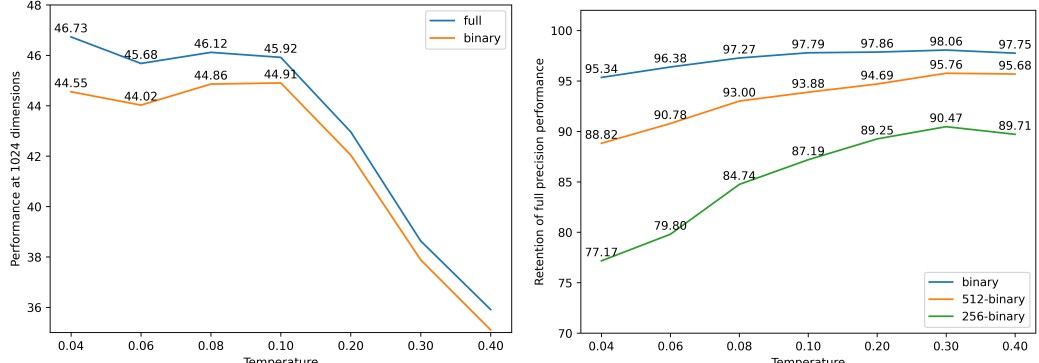

Figure 6: MTEB Retrieval performance when embeddings are *binarized*: Absolute performance (at 1024) and quality retention computed relative to full precision performance (at 256, 512 and 1024). Same as with truncation, peak absolute performance is obtained with the lowest temperature in the un-compressed case; however mid-range values are optimal after binarization due to the increase in quality *retention* associated with larger $\tau$ (right-hand side figure).

Finally, when comparing the two methods of reducing embedding size, binarization is far superior: with $\tau = 0.1$ it achieves a 32x size reduction with 97.8% quality retention versus 4x with 96.4% quality retention when truncating to 256. When the two methods are combined, despite obtaining a reduction of 128x in size, quality drops significantly with a retention of 87% for the same $\tau = 0.1$.

**Matryoshka representation learning (MRL)** The prior experiments tested if larger temperatures lead to embeddings that can be compressed better *for free*, meaning without a training-aware algorithm. We perform an additional set of experiments, this time using MRL (Kusupati et al., 2024), which is a simple method that has been shown to work well in practice. MRL uses a loss function that additionally optimizes for smaller vectors obtained by truncating the original vectors:

$$L_{\mathrm{MRL}}(f, x, \tau) = \Sigma_{i=1}^{k} \lambda_i L_{\mathrm{InfoNCE}}(f_{[1:d_i]}, x, \tau) \tag{5}$$

where $f_{[1:d_i]}$ truncates vectors to dimensions $d_i \leq d$, the original dimensionality of the space. We set $k = 3$ and $d_i$ to $[256, 512, 1024]$, and we use uniform $\lambda_i$ weights.

Results are shown in Table 1. We first observe that MRL training improves over the best retrieval results even for full size embeddings. As expected it also improves both absolute performance and quality retention at 256; this effect is more pronounced for smaller $\tau$, but it is present across the entire temperature range. Interestingly, MRL training also improves *binary* performance, but only at small $\tau$ values, taking quality retention from 95.3 to 96.1 for $\tau = 0.04$. Finally, it is interesting to observe that MRL training lowers the intrinsic dimensionality of the space across all temperature values. This further strengthens the observed correlation between lower intrinsic dimensionality and improved truncated representations.

Overall, the main findings of this section can be summarized as:

1. Larger temperature values in InfoNCE lead to lower intrinsic dimensionality of the resulting space; however, the mechanism through which this happens is still not clear. Larger temperatures lead to all negative samples contributing similarly to the loss. It may be the case that multiple (and heterogeneous) negatives lead to a larger number of smaller updates in the embedding space, subsequently resulting in redundant encoding of information across dimensions. In future work, we plan to investigate this hypothesis and experiment with different batch sizes and methods for negatives selection.

2. Lower intrinsic dimensionality is correlated with better quality retention when embeddings are compressed. This holds both for random feature selection and for binarization.

3. MRL training, which promotes embeddings that can be truncated with minimal performance loss, further reduces the intrinsic dimensionality of the resulting space. This happens across different temperature values and leads to better quality retention when compressing with both truncation *and binarization*.

Table 1: MTEB Retrieval performance across different $\tau$ values for InfoNCE (Baseline) and MRL objectives. Full performance (first row) stands for performance at full dimensionality (1024) and at full precision. The rest of the results reduce embedding size by truncating to 256 (while keeping full precision) and binarizing (while keeping 1024 dimensions).

|  | **Baseline** | | | **MRL** | | |
|  | Temperature | | | Temperature | | |
|  | .04 | 0.1 | 0.3 | .04 | 0.1 | 0.3 |
| Perf. full | 46.7 | 45.9 | 38.6 | **47.2** | 45.4 | 38.8 |
| Perf. at 256 | 43.6 | 44.2 | 38.2 | **45.2** | 44.4 | 38.5 |
| Perf. at binary | 44.6 | 44.9 | 37.9 | **45.3** | 44.3 | 38.1 |
| Perf. 256/1024 | 93.2 | 96.4 | 98.8 | 95.8 | 97.8 | **99.0** |
| Perf. binary/full prec. | 95.3 | 97.8 | **98.1** | 96.1 | 97.7 | 98.0 |
| Intrinsic dim. | 654 | 623 | 430 | 626 | 616 | **396** |

**Related work on embedding compression** There is a large body of literature on dimensionality reduction and quantization for embeddings. Most recent work proposes training-aware methods, where losses or architectures are modified such that either 1) compressed embeddings or 2) both full and compressed embeddings are predictive of the task at hand (Kusupati et al., 2024; Bai et al., 2021; Zafrir et al., 2019; Lin et al., 2017; Yamada et al., 2021; Izacard et al., 2020; Chen et al., 2022). In the case of quantization, most studies focus on quantizing model *weights*, which also reduces inference time. It is difficult to assess where our results fit within this body of work, due to differences in experimental settings, benchmarking, and goals.

More relevant to our experiments is the report by Shakir et al. (2024) which evaluates open source and proprietary models when vector truncation and binarization are used to reduce the size of embeddings post-training. Specifically, the authors highlight the cost of storing large embeddings and test several models after binarization. They show that some models can achieve 92% performance retention with binarization, and 96% when binarization and reranking is applied. These are lower than the retention scores we obtain, indicating that other differences, most likely in the way negatives are selected, further impact this property of the embedding spaces. When truncation is applied, results vary depending on the model used and the compression ratio obtained. For example, the authors report 90% with 6x compression for one model and 93.1% with 12x compression with another, which places our results in a similar range: we obtain 94% retention with 4x reduction at $\tau = 0.04$.

## 5 MULTIPLE TEMPERATURES IN TRAINING

The previous section shows that larger temperatures lead to embeddings that can be more efficiently compressed. However, there is still a trade-off between this desirable property and performance in absolute terms. In this section, we investigate the use of multiple temperature values during training in order to obtain a better trade-off between the two. We opt for aggregating loss terms which use individual temperature values, which is simpler than temperature schedules (employed for example in Kukleva et al., 2023).

**Plain temperature aggregation** For this setup, we replace the standard InfoNCE loss with a sum over losses, each using individual temperature values.

$$L_{\text{TempAgg}}(f, x, \tau) = \Sigma_{t=1}^{T} w_t L_{\text{InfoNCE}}(f, x, \tau_t) \tag{6}$$

where we use $\tau$ to stand for a **vector** of temperature values. In our experiments, we sum over $T = 3$ temperature values $\tau \in [0.03, 0.06, 0.1]$, which cover the best range of values observed in previous sections. We set $w_i = 1.0$.

**MRL temperature aggregation** Previous experiments showed that MRL can further improve performance at reduced dimensions with no degradation at full dimensionality. We test a loss that combines the two; it differs from the original MRL loss by allowing for temperature aggregation at all dimensions in training:

$$L_{\text{TempAggMRL}}(f, x, \tau) = \Sigma_{i=1}^{k} \lambda_i L_{\text{TempAgg}}(f_{[1:d_i]}, x, \tau) \tag{7}$$

Table 2: MTEB Retrieval (Ret.) and Clustering (Clust.) performance across different models. "Full" stands for performance at full dimensionality (1024) and at full precision. The rest of the results report retrieval when: 1) truncating to 256 while keeping full precision and 2) binarizing both with re-ranking (bin re-rnk) and without, while keeping 1024 dimensions.

|  | $\tau = 0.04$ | $\tau = 0.1$ | TempAgg | TempAggMRL | TempSpecMRL |
|---|---|---|---|---|---|
| Ret. (full) | **46.7** | 45.9 | 46.4 | 46.6 | 46.0 |
| Clust. (full) | 34.1 | **37.7** | 36.4 | 36.9 | 37.4 |
| Ret. 256 | 43.6 | 44.2 | 44.6 | **45.1** | 44.6 |
| Ret. bin | 44.6 | 44.9 | 45.2 | **45.2** | 44.8 |
| Ret. bin re-rnk | 45.6 | 45.6 | 45.9 | **46.1** | 45.5 |
| Ret. 256/1024 | 93.2 | 96.4 | 96.2 | 96.9 | **97.1** |
| Ret. bin/full prec | 95.3 | 97.8 | **97.4** | 97.1 | **97.4** |
| Ret. bin re-rnk /full prec | 97.6 | **99.0** | 98.9 | 98.9 | **99.0** |
| Intrinsic dim. | 654 | 623 | 629 | 623 | **617** |

In this setup we use the same temperature range as above, which we apply at 256, 512 and 1024 dimensions, using uniform $\lambda_i = 1.0$ weights.

**Temperature specialization** We implement a third setup which uses the observation that lower temperatures are more beneficial for retrieval. Since reduced embeddings are mostly crucial in retrieval applications, we perform MRL training using a small $\tau$ at 256 and a larger $\tau$ at 1024. The small temperature will promote good retrieval performance at 256 while the larger value at 1024 will provide a temperature aggregation effect (note that a loss applied at 1024 will impact the first 256 dimensions, but not viceversa):

$$L_{\text{TempSpec}}(f, x, \tau) = \Sigma_{i=1}^{k} \lambda_i L_{\text{InfoNCE}}(f_{[1:d_i]}, x, \tau_i) \tag{8}$$

We set $\tau = \{256 : 0.03, 512 : 0.06, 1024 : 0.1\}$. In this last set of experiments, we also add re-ranking for binary embeddings, which is an effective method to improve performance in the binarization setup with minimal speed impact and no vector storage impact. This consists of comparing binary passage vectors with *full-precision query* vectors (Yamada et al., 2021). Specifically, we retrieve the top 100 passages using binary representations and then re-rank them using the full-precision query.

Results are shown in Table 2. We compare the aggregation methods with the baseline InfoNCE models using the best $\tau$ for retrieval (0.04) as well as the best $\tau$ across both retrieval at full size and compressed retrieval (0.1). Both TempAgg and TempAggMRL reach performance that is very close the best retrieval model while improving on clustering by a large margin. In parallel, the quality retention after compression also increases: from 95% to over 97% for binarization and from 93% to 96% for truncation to 256. Among these, TempSpecMRL is the best model w.r.t. quality retention. Best binarization results are obtained when adding the re-ranking stage: for several aggregation models, retention reaches an impressive 99% while reducing the size by a factor of 32x.

Finally, the intrinsic dimensionality of the resulting spaces is similar for all aggregation models and is lower than that of the $\tau = 0.04$ models. This is in line with previous observations that intrinsic dimensionality correlates with quality retention when compressing.

## 6 FUTURE WORK

In future work, we plan to further explore the intrinsic dimensionality of embeddings spaces, and the subsequent relationship with compression under different training regimes. Additionally, an interesting question that arises from this work is why binarization (and presumably quantization in general) is a better method for flexible compression than dimensionality reduction. In our experiments, both methods shows similar quality retention across different settings, albeit with different compression ratios (32x and 4x for binarization and truncation, respectively). We will further explore whether this observation can be applied not only to obtain flexible embeddings, but also to improve performance overall as a form of over-parametrization in training.

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

# A FULL TRAINING SETTINGS

Figure 7 shows the training parameters used to train the models. Additionally, note that we assign weights to the negative samples based on similarity to the anchor, following Dong et al. (2024).

```
--max_seq_length 1024
--max_steps 3000
--warmup_steps 58
--base_global_batch_size 4096
--weight_decay 0.1
--base_learning_rate 5e-06
--lr_min_ratio 1e-01
--base_max_steps 3000
--lr_scheduler_type cosine
--gradient_clip_val 1.0
--optimizer FusedAdam
```

Figure 7: Additional training parameters. We use in-batch negatives with a batch size of 256 and homogenous sampling, meaning that the negative sample are drawn from the same training set. All models are tested after 2000 training steps.

Table 3 lists the datasets used in the contrastive training stage.

Table 3: Datasets used in the contrastive training stage. MsMarco is downsampled to the size listed in the table. MIRACL and MrTydi are multilingual while the others are En only. References are given in the main text.

| Dataset | Size (in k entries) |
|---|---|
| MsMarco | 1,000k |
| NQ | 150k |
| NLI | 247k |
| HotPotQA | 90k |
| FEVER | 108k |
| MIRACL | 600k |
| MrTyDi | 40k |

# B RESULTS ON OTHER TASKS

The main results in section 3.2 focus on the effect of temperature on retrieval and clustering tasks (see Figure 2). In this section we discuss the performance and compression rate of other MTEB tasks.

Figure 8 shows performance (on the MTEB benchmark) as a function of temperature for the remaining MTEB tasks. Reranking performance is not affected by temperature. Performance on other tasks shows a similar but much smaller effect to what we observed in retrieval, with higher temperature values leading to lower performance.

In Figure 9, we show retention of full-precision performance when truncating to 256 dimensions at different temperature values. Overall, we see a similar pattern to what we observed in Figure 4 for retrieval, with higher temperature generally leading to higher performance retention, although this trend is particularly pronounced for clustering.

# C RESULTS WITH ADDITIONAL TRAINING SETTINGS

In this section, we vary the training settings to see whether our results generalize to different settings.

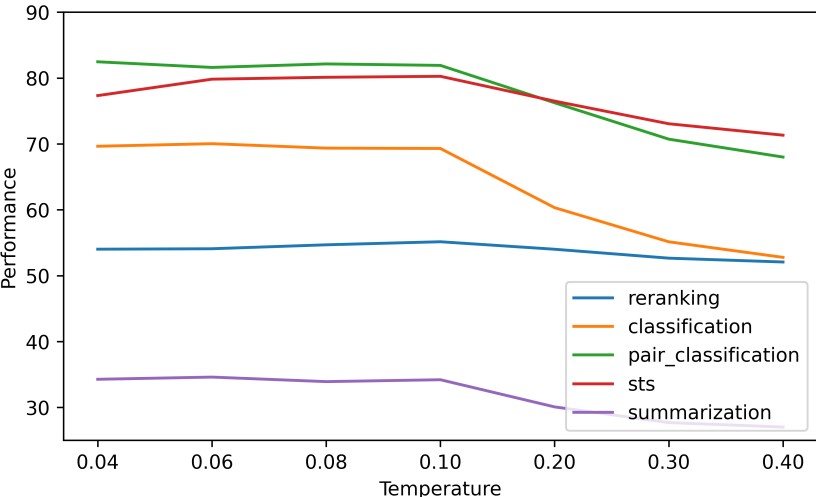

Figure 8: MTEB performance as a function of the temperature value used in the contrastive learning training stage.

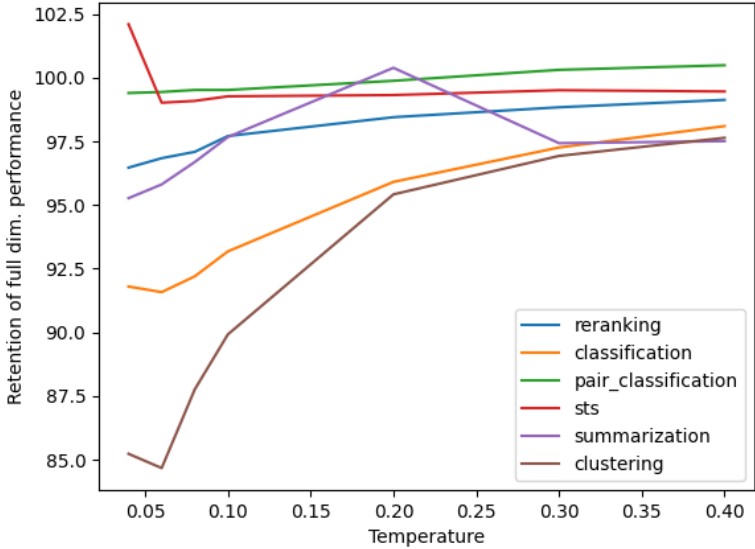

Figure 9: Retention of overall MTEB performance when truncating to 256 dimensions as a function of temperature value used in the contrastive learning training stage.

## C.1 SAMPLING STRATEGY

By default in our experiments, we use homogeneous sampling (i.e., sampling negative examples from the same dataset as the positives) to increase the difficulty of the negatives. Here, we compare homogeneous vs. heterogeneous sampling to see the effect of easier negatives on our experiments. Given that with lower temperatures, more difficult negatives receive a higher weight, we hypothesize that hetereogeneous sampling could have a similar effect to using higher temperatures.

Table 4 compares homogeneous and heterogeneous sampling results. We find that heterogeneous sampling yields worse results than homogeneous sampling, particularly for retrieval. In addition, heterogeneous sampling slightly increases retention of retrieval performance at lower values of $\tau$.

Table 4: MTEB retrieval (Ret.) and Clustering (Clust.) performance with homogeneous (default) and heterogeneous sampling. "Full" stands for performance at full (1024) dimensionality and full precision. The remaining results report retention of retrieval performance when: 1) truncating to 256 while keeping full precision and 2) binarizing without re-ranking while keeping 1024 dimensions

| sampling | $\tau = 0.04$ | | $\tau = 0.3$ | |
|---|---|---|---|---|
| | homogeneous | heterogeneous | homogeneous | heterogeneous |
| Clust. (full) | 34.14 | 33.81 | 39.24 | 38.94 |
| Ret. (full) | 46.73 | 45.70 | 38.64 | 35.80 |
| Ret. 256/1024 | 93.10 | 93.46 | 98.80 | 98.59 |
| Ret. bin/full prec | 95.34 | 95.53 | 98.06 | 97.20 |

Table 5: MTEB retrieval (Ret.) and Clustering (Clust.) performance with different numbers of training steps. "Full" stands for performance at full (1024) dimensionality and full precision. The remaining results report retention of retrieval performance when: 1) truncating to 256 while keeping full precision and 2) binarizing without re-ranking while keeping 1024 dimensions

| steps | $\tau = 0.04$ | | | | $\tau = 0.3$ | | | |
|---|---|---|---|---|---|---|---|---|
| | 1k | 2k | 3k | 4k | 1k | 2k | 3k | 4k |
| Clust. (full) | 34.57 | 34.14 | 33.93 | 34.69 | 37.42 | 39.24 | 37.85 | 37.77 |
| Ret. (full) | 46.05 | 46.73 | 46.55 | 46.69 | 37.53 | 38.64 | 38.36 | 38.33 |
| Ret. 256/1024 | 93.86 | 93.10 | 94.14 | 94.43 | 98.89 | 98.8 | 98.77 | 98.61 |
| Ret. bin/full prec | 95.62 | 95.32 | 94.96 | 95.79 | 97.90 | 98.14 | 97.70 | 97.88 |

However, given that the absolute performance is worse than with homogeneous sampling, this is not useful in practice. Overall, the major patterns (improved clustering and compression at higher $\tau$, improved retrieval at lower $\tau$) remain the same regardless of the sampling method.

## C.2  NUMBER OF TRAINING STEPS

By default, we train our models with 2000 training steps. In Table 5, we report retrieval and clustering performance as well as retention of retrieval performance after truncating or binarizing when training for 1000, 2000, 3000, and 4000 steps. We see no major differences in either full or compressed performance for different number of steps, and the major findings of the paper remain the same regardless of the number of steps. This indicates that our findings are not an artifact of the stage of training.

