# OpenReview forum: "Effective post-training embedding compression via temperature control in contrastive training"
_ICLR.cc/2025/Conference — ICLR 2025 Spotlight_

### Official Review · Reviewer_XBz5 · 2024-10-29

**Soundness:** 3
**Presentation:** 2
**Contribution:** 2
**Rating:** 6
**Confidence:** 2

**Summary:**

This paper investigates the role of the temperature parameter in contrastive training for text embeddings, focusing on its impact on the intrinsic dimensionality of the embedding spaces obtained. The study finds that lower intrinsic dimensionality is correlated with effective compression of embeddings, highlighting the importance of the temperature parameter in embedding compression. This research provides new insights into how the temperature parameter in contrastive learning affects embedding quality and compressibility.

**Strengths:**

The strengths of this paper are summarized below：
(1) The paper presents a novel investigation into the impact of temperature in contrastive training on embedding compression, exploring the relationship between intrinsic dimensionality and post-training compression techniques.
(2)The research demonstrates high-quality experimental design, with comprehensive tests across various temperature values and compression methods, providing robust evidence for their hypotheses.

**Weaknesses:**

There is a substantive assessment of the potential weaknesses of the paper.
(1) The paper may lack significant novelty, as it appears to be building upon existing work in contrastive learning and embedding compression techniques.
(2) The paper's focus on post-training transformation and temperature aggregation methods may be too narrow, potentially overlooking other important factors in embedding quality and compression.
(3) While the paper claims practical implications, it may not sufficiently demonstrate how its findings can be effectively implemented.

**Questions:**

Here are some questions and suggestions for the authors:
（1）Please provide detailed information about the computational costs and resource requirements of the proposed method, especially regarding the efficiency improvements compared to the uncompressed model.
（2）Please clarify the innovations of the proposed method in relation to existing techniques, particularly regarding the impact of the temperature parameter on embedding compression.
（3）It is advisable to revise the visual content and tables in the paper, such as Table 1 and Table 2.

---

### Official Review · Reviewer_11ep · 2024-10-31

**Soundness:** 3
**Presentation:** 3
**Contribution:** 2
**Rating:** 8
**Confidence:** 2

**Summary:**

The paper explores the effect of temperature in contrastive learning for the efficient compression of embeddings. One of the main findings of the paper is that controlling the temperature impacts the intrinsic dimensionality of the embeddings. At the same time, they found that lower temperatures enhance retrieval performance while higher temperatures improve the clustering performance.

They explore how the temperature impacts compression performance by studying two methods of compression, namely truncation and quantization via binarization. The work suggests that we can find an optimal balance between retrieval and clustering performance by tuning the temperature.

**Strengths:**

1) The paper studies in-depth the influence of the temperature during contrastive training on the embedding compression and shows that this is a crucial ingredient to optimality of the resulting retrieval and clustering tasks. It clearly provides a guideline of the trade-off between memory efficiency and task-specific performance by tuning the temperature.

2) This difference in absolute performance and quality retention across temperatures underlines the crucial role temperature plays in embedding quality. While higher temperatures retain compression better, low temperatures ensure higher retrieval accuracy, and this forms the practical basis for a variety of applications.

3) Multi-temperature training strategies allow a good balance between retrieval accuracy and high compression retention, hence underlining the specialization of temperature as an effective way to optimize embeddings for multiple objectives in real-world tasks.

**Weaknesses:**

1) While the study indeed manages to adapt insights on temperature from the vision domain effectively, it is not introducing a fundamentally new theory and limits novelty in its contributions. The findings extend previous knowledge from vision papers without anything particularly new in terms of mechanisms that are specific to the case of text embeddings.

2) It would be helpful if the paper is broadened by exploring other factors that influence embedding compression and performance, such as alternative sampling techniques for negative examples.

3) Limited variation in the temperature values used in experiments. The range of temperature values in the experiments is quite narrow, with further gains/losses possibly missed by not considering a much wider range of configurations. Increasing this range or adaptive adjustment of temperature would reveal more about how best to trade off compression quality and retrieval performance across different tasks.

**Questions:**

The weaknesses mentioned above.

As suggestions, the labels for Figure 3 are hard to read in the first 3 images, they should be made bigger.

---

### Official Review · Reviewer_Gw7k · 2024-11-02

**Soundness:** 3
**Presentation:** 3
**Contribution:** 3
**Rating:** 8
**Confidence:** 4

**Summary:**

The paper focus on analysis of the temperature impact on embeddings' properties when training using contrastive loss. Authors show the trade-off between retrieval and clustering tasks, they attribute to the intrinsic dimensionality. Low intrinsic dimensionality (associated with the large temperature) and high intrinsic dimensionality (corresponds to the low temperature) behaves differently under the compression. The former retains up to 99% of the performance when doing truncation or binarization, while for the latter retention is at most 97.5%.

**Strengths:**

- Well written; clear; easy to follow
- Important analysis work on the embeddings that can benefit different areas of ML
- Interesting connection to embeddings compression

**Weaknesses:**

- There is no comparison to other pre-trained models. So the studied effect can be the property of the model/data combination

**Questions:**

For authors: it was a pleasure to read your work. I have a few questions/suggestions.

- When your report the retrieval/clustering results, are these average results across all tasks?
- I am curious why there is no investigation on quantisation for clustering. Does that make any sense?
- While retention % is higher with the higher temperature, the  absolute quality decreases significantly on retrieval. It took me a while to understand what is happening in Figure 5 and 6, so I suggest you either point it out clearly, or change visualisation method to avoid confusion

---

### Official Review · Reviewer_dzd1 · 2024-11-04

**Soundness:** 4
**Presentation:** 3
**Contribution:** 3
**Rating:** 8
**Confidence:** 4

**Summary:**

This paper researchers the effect of temperature in contrastive learning, specifically its effect on downstream performance as well as compressibility and intrinsic dimension. The idea of "flexible compression" is useful and novel as well. The experiments are intuitive and provide practical insight into why and how embeddings-training works under different temperatures.

**Strengths:**

- Basic research into embedding spaces is important and underexplored. Temperature is a similarly crucial and "taken-for-granted" value for practitioners training models with contrastive loss.
- The result that temperature decreases retrieval performance and increases clustering performance is interesting and useful, since many researchers simply average the performance of both tasks
- Multi-temperature optimization is novel and appears useful (multi-temperature trained models improve over single-temperature ones)

**Weaknesses:**

- Findings are specific to training text embedding models for the zero-shot MTEB benchmark. Insights aren't general to contrastive loss (i.e. theoretical) or multimodal or vision models, even though those models also use the temperature parameter.
- Multi-temperature optimization isn't super principled
- No discussion of learnable temperature value (as used in the original CLIP paper, for example)

**Questions:**

- Would learnable temperature improve results?
- Why only optimize over three temperature values?
- Do you expect these results to generalize to other domains, e.g. vision?
- Is there any connection between your observations about the effect of temperature and the research into temperature values for the more general language modeling task?

---

### Meta-Review · Area_Chair_cf13 · 2024-12-11

**Metareview:**

(a) Scientific Claims and Findings:
This paper investigates the effect of temperature in contrastive learning for text embeddings, specifically its impact on downstream performance, compressibility, and intrinsic dimension.  The authors claim that controlling temperature affects the intrinsic dimensionality of the embeddings, leading to a trade-off between retrieval and clustering performance.  Lower temperatures enhance retrieval, while higher temperatures improve clustering.  The paper also explores how temperature impacts compression performance, suggesting that an optimal balance between retrieval and clustering performance can be found by tuning the temperature.

(b) Strengths:
Provides a deep dive into the impact of temperature on text embedding models, which is an important and underexplored area.

Offers practical insights into why and how embedding training works under different temperatures.

The multi-temperature optimization approach is novel and shows improvement over single-temperature models.

(c) Weaknesses:
Findings are specific to training text embedding models and may not generalize to other contrastive learning scenarios.

The multi-temperature optimization approach lacks strong theoretical grounding.

The paper doesn't discuss the use of learnable temperature values.

Limited variation in temperature values used in the experiments.

The paper could benefit from exploring other factors that influence embedding compression, such as alternative sampling techniques.

(d) Reasons for Acceptance:
The paper makes a valuable contribution to the understanding of the role of temperature in contrastive learning for text embeddings. The findings are interesting and provide practical insights for practitioners. While there are limitations in terms of scope and theoretical grounding, the novelty of the multi-temperature optimization approach and the potential for future research in this area warrant acceptance.

**Additional Comments On Reviewer Discussion:**

Summary of Rebuttal Discussion and Changes:
Reviewers raised several questions and concerns, including the lack of generalization to other domains, the limited exploration of temperature values, and the absence of a comparison to learnable temperature values.  The authors responded by acknowledging the limitations of their work and suggesting future research directions.  They also added additional results in the appendix to further support their findings.

Specific Points Raised and How They Were Addressed:
Generalization to other domains: The authors acknowledged that their findings are specific to text embedding models and expressed hope that their work would lead to future theoretical insights and applications in other domains.

Limited exploration of temperature values: The authors defended their choice of three temperature values, stating that they were chosen to cover the full range of good values.  They also mentioned conducting additional experiments with a lower temperature value, which resulted in poor performance.

Learnable temperature values: The authors acknowledged that learnable temperature values were not investigated in the paper but suggested that it could be a potential area for future work.

Weighing in Each Point in the Final Decision:
The authors' responses during the rebuttal period adequately addressed the concerns raised by the reviewers. While the limitations in scope and theoretical grounding remain, the authors' willingness to acknowledge these limitations and suggest future research directions is encouraging. The additional results in the appendix further strengthen the paper's contributions. Overall, the paper's strengths outweigh its weaknesses, and the findings are sufficiently interesting and relevant to warrant acceptance.

---

### Decision · Program_Chairs · 2025-01-22

Accept (Spotlight)